# Molecular Mechanisms and Therapeutic Strategies to Overcome Resistance to Endocrine Therapy and CDK4/6 Inhibitors in Advanced ER+/HER2− Breast Cancer

**DOI:** 10.3390/ijms26073438

**Published:** 2025-04-07

**Authors:** Paola Ferrari, Maria Luisa Schiavone, Cristian Scatena, Andrea Nicolini

**Affiliations:** 1Department of Oncology, Pisa University Hospital, Via Roma 57, 56126 Pisa, Italy; cristian.scatena@unipi.it (C.S.); andrea.nicolini@med.unipi.it (A.N.); 2Division of Pathology, Department of Translational Research and New Technologies in Medicine and Surgery, University of Pisa, 56126 Pisa, Italy; marialuisa.schiavone@phd.unipi.it

**Keywords:** advanced ER+/HER2− breast cancer, endocrine therapy, CDK4/6 inhibitors, endocrine resistance

## Abstract

Approximately 70–80% of breast cancers are estrogen receptor-positive (ER+), with 65% of these cases also being progesterone receptor-positive (ER+PR+). In most cases of ER+ advanced breast cancer, endocrine therapy (ET) serves as the first-line treatment, utilizing various drugs that inhibit ER signaling. These include tamoxifen, a selective estrogen receptor modulator (SERM); fulvestrant, a selective estrogen receptor degrader (SERD); and aromatase inhibitors (AIs), which block estrogen synthesis. However, intrinsic or acquired hormone resistance eventually develops, leading to disease progression. The combination of ET with cyclin-dependent kinase 4 and 6 inhibitors (CDK4/6is) has been shown to significantly increase progression-free survival (PFS) and, in some cases, overall survival (OS). CDK4/6is works by arresting the cell cycle in the G1 phase, preventing DNA synthesis, and enhancing the efficacy of ET. This review highlights the key mechanisms of resistance to ET, whether used alone or in combination with biological agents, as well as emerging therapeutic strategies aimed at overcoming resistance. Addressing ET resistance remains a work in progress, and in the near future, better patient selection for different therapeutic approaches is expected through the identification of more precise biological and genetic markers. In particular, liquid biopsy may provide a real-time portrait of the disease, offering insights into mechanisms driving ET resistance and cancer progression.

## 1. Introduction

Approximately 70–80% of breast cancers are ER+ [1], with 65% of these also being ER+PR+ [2]. In most cases of advanced ER+ disease, endocrine therapy (ET) serves as the primary treatment option [3,4]. ET encompasses various drugs that function by inhibiting ER signaling, including tamoxifen, a selective estrogen receptor modulator (SERM); fulvestrant, a selective estrogen receptor degrader (SERD); or aromatase inhibitors (AIs), which reduce estrogen production [3,4]. However, resistance to hormone therapy eventually develops, leading to disease progression [5]. Primary endocrine resistance refers to disease relapse within 24 months of initiating adjuvant ET or progression within the first six months of first-line ET for advanced or relapsed breast cancer. Disease relapse following the first 24 months of adjuvant ET, or within 12 months following the end of adjuvant ET, or progression more than 6 months from starting ET for metastatic disease is called secondary or acquired resistance [6]. The addition of cyclin-dependent kinase 4 and 6 inhibitors (CDK4/6is) to ET has recently been associated with improved progression-free survival (PFS) and, in some cases, overall survival (OS) in patients with invasive ER+ breast cancer. By blocking the cell cycle in the G1 phase and halting DNA synthesis, CDK4/6is acts synergistically with ET [7]. This review focuses on the mechanisms of resistance to ET, whether used alone or in combination with biological agents, and on new drugs/strategies currently in use or under investigation to overcome it.

## 2. Mechanisms of Resistance to ET

A few various modalities for the arising lack of response to ET have been described [8]. They comprehend genetic aberrations (mutations, fusions, amplifications) in the ligand-binding domain of the *ESR1* gene, which encodes ER-alpha, mechanisms that refer to regulators of the ER pathway, the PI3K/AKT/mTOR or other signaling cascades, metabolic reprogramming, and further processes.

### 2.1. ESR1 Genetic Alterations

*ESR1* mutations (ESR1m) are among the most prevalent genetic alterations contributing to ET failure [9]. During ET, ER+ breast cancer patients can develop specific mutations, such as Y537S in ERα, which drive uncontrolled cell growth and endocrine-resistant metastatic disease [10]. Proteomic analysis of breast cancer cell lines harboring Y537N and Y537S ER mutations has revealed a significant upregulation of immune-related pathways, along with increased proliferation signaling and activation of key kinases, particularly mTOR and CDKs. This suggests that *ESR1* mutations are linked to enhanced cyclin-dependent kinase signaling [11]. Chromosomal abnormalities of the *ESR1* gene identified in ET-resistant breast cancer include gene fusions within the same chromosome or with oncogenes on different chromosomes [12,13]. Additionally, increased *ESR1* copy number and elevated ER-alpha expression levels have been observed in 1% to 37% of endocrine-refractory breast cancers [14,15]. However, the link between *ESR1* amplifications and ET resistance, particularly reduced sensitivity to tamoxifen, remains a subject of ongoing debate [16,17].

### 2.2. Regulators of the ERα Pathway

Alterations in co-factors, chromatin modifiers, or miRNAs may contribute to endocrine resistance. A study on luminal breast cancers found that cancer-associated fibroblast (CAF) infiltration near malignant cells was associated with reduced ER-α expression and function. CAFs facilitated estrogen-independent tumor growth by maintaining the expression of genes linked to treatment resistance, basal-like differentiation, and dissemination while suppressing most estrogen-responsive genes. Additionally, genes downregulated by CAFs in cancer cells correlated with lower ET sensitivity, and the transforming growth factor-β (TGF-β) and Janus kinase signaling pathways were identified as potential targets to counteract CAF-driven changes through ER-α modulation [18]. MYSM1, a deubiquitinase highly expressed in breast cancer samples, was shown to regulate ERα activity, and its depletion in xenograft models reduced breast cancer cell proliferation while enhancing sensitivity to anti-estrogen therapies [19]. Furthermore, a bioinformatics analysis of primary N+ breast tumors, validated in a mouse model, identified MNX1 as a potential transcription factor that influences hormone receptor (HR) levels alongside HER2. Macrophages emerged as key components of the tumor microenvironment (TME), contributing to HR downregulation and HER2 upregulation, likely through MNX1 activity [20]. SMAD4 acts as a signal transducer in the transforming growth factor (TGF) superfamily. Some ductal carcinomas have homozygous deletions of SMAD4. In HR+/HER2− breast cancer cells, SMAD4 has been shown to work as an ER transcriptional corepressor and to counteract tumor progression by promoting apoptosis. SMAD4, by a genome-wide CRISPR analysis, was identified to be relevant in controlling 4-hydroxytamoxifen (OHT) sensitivity in T47D cells. Based on clinical findings, it was found that the downregulation of SMAD4, induced by ET, contributed to the rising of acquired resistance by cooperation of ER with ERBB signaling [21].

### 2.3. The PI3K/AKT/mTOR and Other Signaling Pathways

Mutations or amplifications in the *ERBB2* gene are observed in approximately 5% and 2% of the ER+/HER2− relapsed breast cancer population, respectively [22], while nearly half of these cases exhibit increased activity of the PI3K/Akt pathway [23]. *ErbB2/HER2/Neu* amplification reduces the effectiveness of anti-estrogen therapies, primarily by enhancing PI3K/Akt and MAPK signaling, which drive estrogen-independent ER phosphorylation and activation [24]. Additionally, mutations in other components of the receptor tyrosine kinase (RTK)-MAPK pathway, such as *BRAF*, *RAS*, and *MAP2K1*, have been identified in metastatic breast cancers that do not respond to AIs [25]. *PIK3CA* mutations occur in approximately 40% of cases, often in association with endocrine resistance [26,27], while mutations in *AKT1* and *PTEN* are also frequently reported [28]. Upstream signaling through FGFR, EGFR, and other RTKs can lead to hyperactivation of the PI3K/Akt pathway. Furthermore, this pathway is downstream of several growth factors, including insulin-like growth factor-I (IGF-I) and heregulin, both of which utilize it to activate ER and promote estrogen-independent growth [29]. In experimental models of ER+ breast cancer cell lines, prolonged endocrine deprivation led to the emergence of acquired resistance driven by increased PI3K signaling, with a proteomic analysis revealing enhanced phosphorylation of mTOR and Akt [25]. Based on experimental and clinical evidence, combining therapies that target both ER and PI3K/Akt signaling has been suggested as an effective strategy to counteract acquired endocrine resistance [30,31]. Everolimus, an mTOR inhibitor with antitumor activity independent of *PIK3CA* mutation status, is approved in combination with exemestane for the treatment of ER+ metastatic breast cancer [32]. The PI3K/Akt pathway drives endocrine resistance through ligand-independent ER activation, particularly via S6 kinase, a substrate of mTOR complex 1, which phosphorylates the activation function domain 1 (AF1) of ER [33,34]. Everolimus overcomes mTORC1 resistance by binding to and allosterically inhibiting it [35]. In patients with *ESR1* mutations who progress after prior AI therapy and who may also exhibit resistance to everolimus, fulvestrant could be a more suitable anti-estrogen partner. The National Comprehensive Cancer Network (NCCN) guidelines recommend fulvestrant or tamoxifen as alternative partners for ET in combination with everolimus [3,4]. Alpelisib, a PI3K inhibitor, has received FDA approval for the treatment of *PIK3CA*-mutated ER+ metastatic breast cancer when used in combination with fulvestrant [3,4,31]. This approval applies to patients who have not previously been exposed to CDK4/6, PI3K, Akt, or mTOR inhibitors [26]. Inhibitors targeting the PI3K/Akt/mTOR pathway enhance the efficacy of anti-estrogen therapies due to the extensive crosstalk between these pathways and ER [36]. In endocrine-resistant breast cancer cells, cGAS-STING signaling is significantly reduced compared to endocrine-sensitive cells. This decrease appears to be driven by the hyperactivation of AKT1 kinase, leading to a positive feedback loop that further sustains AKT1 hyperactivation and promotes endocrine resistance. Disrupting this cycle by combining an AKT1 inhibitor with a STING agonist has been shown to reduce tumor proliferation in mouse models of endocrine-resistant breast cancer [37]. Amplifications of RTKs such as EGFR (epidermal growth factor receptor) and FGFR1 (fibroblast growth factor receptor 1) are frequently observed in breast cancers that do not respond to ET [8]. FGFR1 overexpression often occurs in aggressive Luminal B-like tumors and correlates with poor outcomes and resistance to therapy. The amplification of *FGFR1* alongside other oncogenes suggests that it is not the sole oncogenic driver. While *FGFR2* amplification is less common, it plays a critical role in regulating hormone receptors, promoting tumor growth, and driving therapy resistance. *FGFR3* and *FGFR4* can also contribute to endocrine resistance through various mechanisms, including activation of the PI3K/AKT/mTOR and RAS/RAF/MEK/ERK signaling pathways [38]. Neurofibromatosis type 1 (*NF1*) is a tumor suppressor gene, and approximately 27% of breast cancers exhibit NF1 alterations. The loss or dysfunction of NF1 leads to RAS pathway hyperactivation and cyclin D1 de-repression, resulting in uncontrolled cell growth and reduced sensitivity to ET [39,40]. Breast cancer is characterized by a highly inflammatory tumor microenvironment (TME). The NF-κB pathway plays a key role in linking inflammation to cancer progression, as well as promoting tumor proliferation and therapy resistance. Chemotherapy, targeted therapy, ET, and radiotherapy may contribute to resistance through NF-κB activation. However, the use of NF-κB inhibitors in combination with tamoxifen has been shown to significantly enhance the response of breast cancer cells to tamoxifen treatment [41].

### 2.4. Metabolic Reprogramming

Some studies have identified a link between metabolic reprogramming and response to ET. Nine metabolic proteins were identified, with phosphomannose mutase 2 (PMM2) emerging as a more promising candidate, while depletion of PMM2 led to the deterioration of ERα Y537S mutant cells, inhibited cell growth, and reduced ERα signaling. In particular, reducing PMM2 levels re-sensitized ERα Y537S-expressing cells to some drugs given for endocrine treatment and to CDK4/CDK6is. Moreover, a PMM2 decrease lowered FOXA1, a protein that is relevant in regulating ERα. Overall, this suggests that PMM2 in relapsed breast cancer with the ERα Y537S variant is a suitable target for treatment [10]. In ER+ breast cancer cells that were maintained long-term without estrogens (LTED cells)—a model commonly used to study resistance to Ais—an increase in intracellular lipid accumulation was found. This metabolic reprogramming was driven by acetyl-CoA carboxylase-1 (ACC1), and inhibiting ACC1 significantly reduced the survival of LTED cells. However, the addition of branched and very long-chain fatty acids reversed the effects of ACC1 inhibition, a phenomenon also observed in patient-derived samples treated with AIs. These findings highlight ACC1 as a potential therapeutic target to prevent the development of estrogen independence in ER+ breast cancers [42]. A study on *NF1*-deficient ER+ breast cancer revealed that NF1 loss drives metabolic reprogramming, leading to impaired oxidative ATP production, increased glutamine flux into the TCA cycle, and an expansion of lipid pools. The researchers concluded that these metabolic changes could create new therapeutic opportunities by combining metabolic and targeted inhibitors [43].

### 2.5. Further Mechanisms

*CYP19A1*, the gene encoding the aromatase enzyme responsible for estrogen production, is frequently overexpressed in endocrine-resistant breast cancer that progresses after AI therapy [44]. Additionally, deletions or mutations in the H3K4 methyltransferase KMT2C (MLL3) histone have been identified in ER+ metastatic breast cancer following AI treatment and are linked to poor PFS [45]. The long non-coding RNA (lncRNA) LINC00152 plays a role in tamoxifen resistance by inhibiting ferroptosis, a type of iron-dependent programmed cell death that is promoted by tamoxifen. Notably, high LINC00152 expression is strongly associated with increased PDE4D, reduced ferroptosis, and worse clinical outcomes. These findings suggest that LINC00152 and its downstream effectors may serve as promising therapeutic targets to improve survival in tamoxifen refractory ER+ breast cancer [46]. A high expression of collagen type XI alpha 1 (COL11A1) has been associated with therapeutic resistance and poor outcomes in breast cancer subjects undergoing tamoxifen. The research evaluated MCF-7/COL11A1 and T47D/COL11A1 cell lines that, unlike the parental MCF-7 and T47D cell lines, exhibited greater resistance to growth inhibition that was induced by 4-OHT, while the knockdown of COL11A1 significantly made these cells sensitive when in vitro and in vivo to 4-OHT. Particularly, in tamoxifen refractory cells, Erα overexpression occurred, likely because of the elevated COL11A1 levels. Additionally, COL11A1 knockdown led to ERα and its downstream target genes’ reduced expression [47]. A meta-analysis in breast cancer confirmed an association between the breast cancer resistance 4 (*BCAR4*) gene and resistance to ET, with BCAR4 expression being clinically significant in both luminal A and B subtypes. Furthermore, a correlation was identified between BCAR4 expression and a lack of response to AIs [48]. In another study, F-box protein 22 (Fbxo22) negativity was linked to ET resistance. Among patients treated with SERMs, those lacking Fbxo22 had worse outcomes, with 10-year OS rates of 81.3% compared to 92.3% (*p* = 0.032). This suggests that Fbxo22 negativity has a significant impact on survival, particularly in patients with invasive ductal carcinoma (IDC) and invasive lobular carcinoma (ILC). The survival disadvantage was especially pronounced in postmenopausal women with ILC or those receiving SERMs [49]. Additionally, another study highlighted the role of RFC3 in the cell cycle, showing that RFC3 overexpression was present in ET-resistant breast cancer cells but absent in parental cells. These findings suggest that RFC3 contributes to ET resistance in breast cancer by promoting cell cycle progression [50].

## 3. CDK4/6is in Association with Endocrine Therapy

The cell-division cycle is regulated by various cyclins and CDKs (Figure 1A,B). Cyclin D1 interacts with CDK4/6 to facilitate progression through the G1 phase, and it is a known target of the PI3K/AKT/mTOR signaling pathway [51]. Given that tumor growth often relies on uncontrolled cell cycle progression, the cyclin D1/CDK4/6 axis has emerged as a key target for cancer therapy, including breast cancer [52,53]. Additionally, CDK4/6 is involved in crosstalk with the ER signaling pathway [54]. Early studies established a strong correlation between ER expression and a response to CDK4/6is [55]. In the PALOMA-2, MONALEESA-2, MONARCH-3, and MONALEESA-7 phase III trials, the addition of CDK4/6is (palbociclib, ribociclib, abemaciclib) to ET significantly improved PFS when used as first-line therapy in both pre- and postmenopausal breast cancer patients [56,57,58,59]. However, a statistically significant overall survival (OS) benefit has only been demonstrated in the MONALEESA-7 and MONALEESA-2 trials, where ribociclib plus anti-estrogens showed prolonged OS in both pre- and postmenopausal breast cancer patients [60,61]. As a result, the current guidelines [3,4] recommend CDK4/6is combined with ET as the standard first-line therapy for HR+/HER2− advanced or relapsed breast cancer. A systematic review of PubMed and Embase (covering studies from February 2015 to March 2022) assessed the cost-effectiveness of CDK4/6is in metastatic breast cancer treatment. The findings revealed that none of the three CDK4/6is provided a positive incremental net benefit (INB) when compared to AIs alone [62]. Furthermore, PFS/OS benefits are often less significant when CDK4/6is are used as second-line therapy, and chemotherapy is preferred for rapid disease control in cases of impending organ failure or a life-threatening visceral crisis. Nevertheless, CDK4/6is combined with fulvestrant has demonstrated efficacy as second-line ET after relapse or progression on an AI [63,64,65,66].

## 4. Resistance to ET and/or CDK4/6is

Several mechanisms contributing to the lack of or reduced response to ET and/or CDK4/6is (CDK4/6is) have been identified. These include genetic aberrations affecting cell cycle regulation, activation of alternative molecular pathways, and alterations in transcriptional and epigenetic regulators [67,68]. Additionally, acquired CDK6 amplification and *c-Myc* oncogene alterations have been implicated. Furthermore, immunological changes within the tumor microenvironment (TME) and continued proliferation despite CDK suppression represent additional potential mechanisms of resistance.

### 4.1. Genetic Alterations Involving Cell Cycle Regulation

Mutational signatures associated with apolipoprotein B mRNA-editing enzyme, catalytic subunit 3 (APOBEC3) enzymes are overexpressed in HR+ breast cancers following therapy compared to untreated cases. These APOBEC3-driven mutations have been independently linked to shorter PFS in patients receiving a combination of anti-estrogen therapy and CDK4/6is (CDK4/6is) for HR+ relapsed breast cancer. Whole-genome sequencing (WGS) of experimental breast cancer models and selected primary and relapsed tumor samples revealed that active APOBEC3 mutations contribute to resistance against endocrine and targeted therapies, primarily through *RB1* mutations, leading to its functional impairment [69]. In one study, the effectiveness of CDK4/6is in relapsed breast cancer with *ESR1* mutations (ESR1m) was assessed using circulating tumor DNA (ctDNA) analysis. The results indicated that ESR1 variants are not associated with pan-CDK4/6i resistance, supporting the hypothesis that combining CDK4/6 inhibition with a SERD may be an effective approach for treating *ESR1*m metastatic breast cancer [70]. Overexpression of the *CCND1* gene, which encodes cyclin D1 (the primary partner of CDK4/6), has been commonly observed in breast cancers resistant to CDK4/6is [71]. Similarly, *CCNE1* gene overexpression, which encodes cyclin E1 (a key cofactor for CDK2, essential for Rb hyperphosphorylation), was correlated with lower response rates to palbociclib in the PALOMA-3 trial [72]. Activating mutations in critical regions, such as the ATP-binding pocket of CDK4/6, have also been implicated in resistance [68]. Additionally, impairing mutations or loss of the *RB1* gene have been reported in breast cancer patients with poor responses to CDK4/6is [73,74]. Tumors with germline (g) *BRCA2* mutations or a homologous recombination deficiency (HRD) associated with g*BRCA2*, as well as baseline loss of heterozygosity (LOH) for *RB1*, exhibited a higher prevalence of RB1 loss-of-function mutations and demonstrated poor outcomes when treated with first-line CDK4/6i combinations. This suggests that using PARP inhibitors prior to initiating CDK4/6i therapy may help intercept RB1-loss-driven resistance pathways and reduce the risk of CDK4/6is resistance development [75]. A study comparing copy number abnormalities (CNAs) in early breast cancers from multiple cohorts to those in metastatic breast cancers identified key genetic alterations associated with ER+/HER2− metastatic disease. Among the 21 most frequently altered genes, focal amplification of *TERT* was significantly linked to worse clinical outcomes. Seven genes were identified in post-treatment biopsies in patients with CDK4/6i treatment and in the available data prior to biopsy collection. Notably, CDK4 overexpression was observed in 9.8% of the pretreated ER+/HER2− relapsed breast cancer patients, compared to just 1.5% in the untreated ER+/HER2− metastatic cohort (*p* = 2.82 × 10^−4^) and in all three early breast cancer cohorts. Furthermore, CDK4 overexpression was associated with poor outcomes in ER+/HER2− early breast cancers [76]. In proliferating cells, the E3 ubiquitin ligase anaphase-promoting complex/cyclosome (APC/C) plays a crucial role in preventing premature entry into the S phase. A study found that the APC/C inhibitor, EMI1, is essential for regulating APC/C function, preventing S phase entry in cells arrested by palbociclib inhibition. This suggests that overexpressing EMI1 cancers are likely to avoid CDK4/6 inhibition, leading to premature and under-licensed S phase entry, which may result in increased genome instability [77].

### 4.2. Activation of Alternative Signaling Pathways

A study on ER+ HER2− relapsed breast cancer patients undergoing ET combined with palbociclib examined a panel of miRNAs. Seven miRNAs showed a significant inverse correlation with PFS, with some being particularly associated with poor outcomes when PFS was less than six months. Multivariate analysis confirmed that miR-378e, miR-99b-5p, and miR-877-5p had a significant and independent impact on PFS. Based on bioinformatics and literature data, PI3K/AKT/mTOR signaling, cell cycle regulators (such as cyclin D1 and CDKN1B), and autophagy were the primary pathways implicated [78]. Resistance to PI3K inhibitors is characterized by persistent Rb phosphorylation, which can be effectively counteracted by combining a CDK inhibitor with a PI3K inhibitor [54]. Regarding the activation of alternative pathways, genetic alterations in *AKT1*, *AURKA*, and *KRAS* have been identified in HR+/HER2− breast cancer that is resistant to CDK4/6is. In particular, activating mutations or overexpression of *AKT1* and *AKT3* have been linked to reduced sensitivity to CDK4/6is [78]. Additionally, alterations in all three *RAS* family members—including *KRASG12D*, *HRASK117R*, and NRAS amplification—were found in HR+/HER2− breast cancers with poor responses to CDK4/6is [78]. Overactivation of FGFR signaling was also observed in CDK4/6is-resistant breast cancer, with *FGFR1/2* overexpression or amplification correlating with reduced sensitivity [79]. Moreover, loss or inactivating mutations in the *FAT1* tumor suppressor gene, which normally inhibits Hippo signaling, led to the enrichment of YAP/TAZ transcription factors on the CDK6 promoter, ultimately causing CDK6 amplification and resistance to CDK4/6is in ER+ breast cancer [80]. A study investigating the role of the G protein-coupled estrogen receptor (GPER) in palbociclib resistance in breast cancer cells revealed that ER-alpha underexpression, combined with GPER overexpression, was driven by EGFR interacting with the GPER promoter region. Additionally, palbociclib was found to activate pro-inflammatory transcriptional events through GPER signaling in cancer-associated fibroblasts (CAFs). Co-culture assays confirmed that GPER signaling reduces palbociclib sensitivity and enhances the functional interaction between breast cancer cells and CAFs, further contributing to resistance [81]. In a separate study, MCF7-FAR and T47D-FAR breast cancer cell lines, which were resistant to fulvestrant and abemaciclib, were developed. These resistant cells exhibited hyperactivation of EGFR, HER2, and AKT signaling. Notably, cetuximab restored tumor sensitivity to fulvestrant and abemaciclib in FAR- and EGFR-overexpressing breast cancer spheroids and xenografts [82]. A proteogenomic analysis of 22 patient-derived xenografts (PDXs) from ER+ breast cancer identified *PKMYT1* as an estradiol (E2)-regulated gene, with intrinsic expression in cases where proliferation was E2-independent. In patient samples, PKMYT1 mRNA overexpression correlated with resistance to ET and CDK4/6is. The PKMYT1 antagonist lunresertib (RP-6306) synergized with gemcitabine, significantly reducing the viability of ER+ breast cancer cells resistant to ET and palbociclib, even in the absence of activated p53 [83]. In another study, tumor and blood samples were collected before initiating combined CDK4/6is and ET, as well as at disease progression. The most frequently acquired alterations included mutations in *PIK3CA* and *TP53*, along with amplification of the pyruvate dehydrogenase lipoamide kinase enzyme 1 (PDK1). Notably, *PIK3CA*-mutated circulating tumor DNA (ctDNA) levels increased 4 to 17 months before disease progression was detectable on imaging, highlighting its potential as an early biomarker for resistance [84]. In another research, the circHIAT1/miR-19a-3p/CADM2 axis affected the epithelial-to-mesenchymal transition (EMT) and resistance to palbociclib with circHIAT1 and CADM2 downregulation in breast cancer tissues and cell lines and miR-19a-3p upregulation. Particularly, palbociclib-resistant breast cancer cells showed similar trends, and overexpressing circHIAT1 in cells restored their sensitivity to palbociclib. Also, the bioactive flavonoid quercetin re-sensitized breast cancer cells to palbociclib. The antitumor effects of quercetin appeared to stem from its capability to govern the circHIAT1/miR-19a-3p/CADM2 axis, mainly by the upregulation of circHIAT1 [85].

### 4.3. Modifications in Transcriptional and Epigenetic Regulators

In breast cancer, the enhanced activity of pro-aggressive transcription factors such as NF-κB, AP-1, and E2F has been associated with resistance to CDK4/6is [68]. A study focusing on the luminal A breast cancer subtype found that CDK4/6is reduces expression of the cystine transporter SLC7A11 by inhibiting SP1 binding to the SLC7A11 promoter region. Both genetic and pharmacological inhibition of SP1 or SLC7A11 in experimental and clinical studies resulted in an improved response to CDK4/6is and a synergistic reduction in cancer proliferation [86]. In another study, the microphthalmia-associated transcription factor (MITF) was found to be activated through O-GlcNAcylation by O-GlcNAc transferase (OGT) in breast cancer cells resistant to palbociclib. Inhibiting either MITF or its O-GlcNAcylation restored sensitivity to palbociclib in these refractory cells. Moreover, MITF activation was also observed in tumor samples from patients resistant to or undergoing palbociclib treatment [87]. Additionally, elevated histone deacetylase (HDAC) activity was linked to tolerance to CDK4/6is, potentially through p21-mediated cell cycle arrest and the activation of survival pathways [88,89]. Additionally, a study found that miR-432 could upregulate CDK6 by inhibiting TGF-β signaling through an SMAD4 decrease, therefore reducing the effectiveness of CDK4/6is [90]. In research, the lncRNA TROJAN led to CDK2 upregulation by linking to NKRF and blocking its inhibition on RELA/p65, so contributing to the decreased response of ER+ breast cancer cells to CDK4/6is [91].

### 4.4. Acquired CDK6 Amplification

CDK6 overexpression has been linked to reduced sensitivity to CDK4/6is in breast cancer [92]. However, it remains unclear whether this is solely due to partial pharmacological inhibition of CDK6 or if kinase-independent activities of CDK6 also play a role [93]. Upregulation of CDK6 may occur as a response to CDK4/6i treatment, and preclinical studies suggest that subsequent suppression of CDK6 can restore treatment response. Recently, a clinical investigation demonstrated an inverse correlation between CDK6 overexpression and PFS in ER+ breast cancer patients undergoing CDK4/6i therapy [94]. Although rare, loss-of-function mutations in the cadherin superfamily member FAT1 [95] have also been associated with CDK4/6is resistance, potentially due to CDK6 overexpression in ER+ breast cancer patients [80]. Notably, *FAT1* loss-of-function or deletion activates the Hippo pathway [96], which plays a crucial role in apoptosis and cell proliferation regulation [97]. Consequently, FAT1 loss leads to increased Yap/Taz transcription factor activity, promoting CDK6 amplification. Patients with biallelic *FAT1* inactivation experience a significantly shorter PFS of 2.4 months, compared to 10.1 months in ER+ breast cancer patients with *FAT1* missense mutations and 11.3 months in patients with wild-type *FAT1* [80]. Furthermore, exosomal miRNA-432-5p contributes to CDK4/6i resistance by promoting CDK6 amplification, reducing SMAD4 expression, and ultimately decreasing G1/S cell cycle arrest [90]. Palbociclib discontinuation has been shown to reduce CDK6, *CCND* (cyclin D1-encoding gene), and miRNA-432-5p levels while simultaneously upregulating RB, suggesting a potential approach to overcoming resistance. Thus, ER+ breast cancer cells refractory to palbociclib can be regulated in the in vitro and in vivo experimental models [90]; this may explain why some ER+ breast cancer subjects who progress during a specific CDK4/6i then might become sensitive to another CDK4/6i.

### 4.5. Oncogene c-Myc Alteration

The *c-Myc* gene, a member of the MYC family, is highly expressed in breast cancer [98]. CDK2, CDK4, and CDK6 play a role in activating Myc, and its levels are elevated in preclinical models exhibiting resistance to CDK4/6is [99]. Consistently, the nextMONARCH-1 clinical trial reported an increase in Myc genomic alterations following treatment with abemaciclib, either as monotherapy or in combination with a nonsteroidal AI [100]. Additionally, activation of the S6K1 kinase has been detected in over 10% of ER+ breast cancer patients, and several studies suggest that elevated S6K1 levels contribute to palbociclib resistance by activating c-Myc signaling, as observed in both experimental models and patient-derived breast cancer samples [101].

### 4.6. Immunological Alterations in Tumor Microenvironment (TME)

Experimental and clinical findings suggest that CDK4/6is promotes several immune effects that, at least in part, account for their efficacy [102]. By this also, it can be inferred that in HR+HER2− breast cancer subjects, not yet well-defined immunological mechanisms may be responsible for low sensitivity to CDK4/6is. Here, some immunological alterations that are more often observed in association with CDK4/6is are considered along with others that are more likely related to the arising of CDK4/6is resistance. A study [103] found that PD-L1 levels are regulated by cyclin D–CDK4 and the cullin 3–SPOP E3 ligase through proteasome-mediated degradation. CDK4/6is enhances PD-L1 expression in vivo by inhibiting cyclin D-CDK4-mediated phosphorylation of speckle-type POZ protein (SPOP), thereby promoting its degradation by the anaphase-promoting complex activator fizzy-related protein homolog (FZR1). Consequently, mutations that inactivate SPOP impair ubiquitination-mediated PD-L1 degradation, leading to increased PD-L1 levels and a reduction in tumor-infiltrating lymphocytes (TILs) in both mouse tumors and primary samples from prostate cancer patients. Notably, CDK4/6is, combined with anti-PD-1 therapy, enhanced tumor sensitivity and significantly improved the overall survival (OS) rates in mouse tumor models. On the other hand, some studies suggest that CDK4/6is may synergize with immune checkpoint blockade agents while also exhibiting suppressive effects on TILs, supporting the clinical use of CDK4/6is alongside anti-PD-1 or anti-CTLA-4 antibodies [104]. Additionally, another experimental study [105] was conducted using parental breast cancer cells and their derivatives that had developed resistance to ET (EndoR). Parental and estrogen deprivation–refractory MCF7 and T47D cells were used to generate palbociclib-resistant breast cancer sublines (PalboR). By a transcriptomic evaluation of these cell lines, an “IFN-related palbociclib-resistance Signature” (IRPS) was identified. In two clinical trials where CDK4/6is plus ET were given in the neoadjuvant setting, IRPS and other IFN-related signatures were overexpressed in a cancer population that was intrinsically insensitive to CDK4/6is. In primary ER+/HER2− tumors, the IRPS score was significantly increased in lum B more than in the lumA subtype and showed a positive correlation with the amplification of immune checkpoints genes, a lack of response to ET, and a poor clinical outcome. In a further experimental study [106], the immunological processes accounting for the arising of insensitivity to CDK4/6is were deeply investigated. An in silico evaluation of the Cancer Genome Atlas (TCGA) was carried out. Moreover, three different cohorts of HR+HER2− breast cancer subjects, including tumor samples collected over time, were examined. An increase in interleukin 17 (IL17) producing gamma/delta T cells occurred in mouse HR+HER2− mammary tumors following CDK4/6 inhibition. In these tumors, circulating IL17 levels led to poor clinical outcomes while inhibiting the gamma/delta TCR through neutralizing IL17 or CCL2, which similarly improved the response to CDK4/6is. These findings were confirmed in patients from the TCGA. In fact, patients with an active IL17 signature showed poor clinical outcomes along with immunosuppression in the TME. In HR+HER2− breast cancer patients, gamma/delta T cell infiltration in tumor tissue samples correlated with the tumor grade, and gamma/delta T cells were placed close to PD-L1+ tumor cells and macrophages.

### 4.7. Proliferation Mechanisms Despite CDK Suppression

The inherent plasticity of cell-division machinery allows breast cancer cells to divide even when all CDKs involved in the interphase are inhibited. Moreover, mice without several CDKs can survive by relying on CDK1. Therefore, it was explored whether breast cells exposed to a combination of CDK2is and CDK4/6is might adopt this specific mechanism [107]. While cells treated with the CDK2i PF3600 [108] may continue to proliferate by utilizing DK1, combining CDK2 and CDK4/6is prevents this compensatory mechanism [109]. In brief, this potential process could be addressed by inhibiting CDK7 (Figure 1), which serves two critical functions: (1) as a cycle-dependent-activating kinase that phosphorylates CDK1, CDK2, CDK4, and CDK6 and (2) acting as an in-between molecule in RNA polymerase II-mediated transcription [110]. Accordingly, specific CDK7is, like samuraciclib, have shown therapeutic efficacy in HR+/HER2− advanced breast cancer populations resistant to CDK4/6is [111]. Furthermore, enhanced cellular growth alone dictates the response to a CDK7i, which helps to explain why there are some cancers sensitive to CDK inhibition more than normally proliferating cells [112]. The main mechanisms of resistance to ET and/or CDK4/6is in advanced ER+/HER2− breast cancer are shown in Figure 2.

## 5. Common Therapeutic Strategies to Overcome Resistance to ET and/or CDK4/6is

After progression on endocrine plus CDK4/6is therapy, there is no established standard systemic treatment. In patients who maintain an ER-dependent signaling, viable options include (a) switching to a different endocrine monotherapy and (b) using ET combined with everolimus (an mTOR inhibitor). Additionally, for patients with somatic *PIK3CA* mutations, ET can be paired with either alpelisib (a PI3K inhibitor) or capivasertib (an AKT inhibitor). In patients who have lost ER-dependent signaling, cytotoxic chemotherapy is an alternative option [3,4,5], while continuing CDK4/6is beyond progression, although experimental, is a further choice. In the principal MONALEESA-2/7, MONARCH-3, and PALOMA-1/2 randomized trials, subjects progressing on first-line CDK4/6is plus ET were given endocrine monotherapy, chemotherapy, or a different CDK4/6is on average in 65%, 44%, or 18%, respectively, of cases, and ET plus mTOR inhibitors in 17% of cases [113]. Anti-estrogen therapy alone obtained a short PFS (less than 3 months) [114].

### 5.1. Fulvestrant, a SERD and Novel Oral SERDs

SERDs are nonsteroidal compounds that function as both competitive ER antagonists and inducers of ER degradation via the proteasome. Fulvestrant, the first SERD approved for ER+ metastatic breast cancer, is administered monthly via im. injection [115]. Fulvestrant as a second-line ET combined with other different drugs like CDK 4/6is, alpelisib, and everolimus has shown to be efficacious in a few randomized trials [26,63,64,66,116,117]. In phase III trials, fulvestrant, given alone after progression on CDK4/6is, was evaluated as a control arm with a limited median PFS (1.9 to 4.7 months, range) [118,119]. *ESR1* mutations occur in 25–40% of patients who were given an AI. In the phase III PADA-1 study, fulvestrant replaced an AI when ctDNA-detected *ESR1* mutations occurred before radiological progression; in patients treated with fulvestrant plus CDK4/6is and with AI plus CDK4/6is, PFS was 11.9 and 5.7 months, respectively [120]. Elacestrant is an oral hybrid SERM/SERD. In the EMERALD phase III trial, participants who previously had undergone one or two lines of ET plus a CDK4/6i and up to one prior line of chemotherapy randomly received either elacestrant or standard of care (SOC) with fulvestrant or an AI. Elacestrant, compared with SOC, significantly increased PFS in subjects with *ESR1* mutations and in the overall population [121], suggesting oral SERD for those maintaining some hormone responsiveness following ET plus CDK4/6is. An update indicated a greater PFS benefit (5.45 and 3.29 months) for patients who had a longer exposure to CDK4/6is, particularly among those with *ESR1* mutations [122]. An interim OS analysis favored elacestrant in the *ESR1*-mutant population, unlike the non-mutant subgroup. Based on this, elacestrant has obtained FDA approval in postmenopausal women or adult men with ER+HER2− *ESR1*-mutated advanced or relapsed breast cancer progressing after at least one line of ET. Elacestrant is under evaluation in combination with other drugs (CDK4/6is, everolimus, alpelisib, samuraciclib) in some clinical trials (NCT05963997, NCT06062498, NCT06382948, and NCT05563220). The ELEVATE (NCT05563220) trial is a phase Ib/II umbrella study, where the clinical performance of elacestrant joined with either alpelisib, capivasertib, everolimus, palbociclib, abemaciclib, or ribociclib is assessed in ER+/HER2− advanced/relapsed breast cancer [123]. ADELA (NCT06382948) is a phase III trial, where elacestrant combined with everolimus is compared to elacestrant alone in the ER+/HER2− advanced breast cancer population with *ESR1*-mutated tumors that have progressed on ET and CDK4/6i. Patients must have received at least one and no more than two lines of ET for advanced breast cancer. Patients receiving a CDK4/6i in the adjuvant setting are eligible only if disease progression is diagnosed after ≥12 months of treatment but <12 months following CDK4/6i treatment completion. Camizestrant (AZD9833) is another oral SERD, a pure antagonist of ER-alpha. Camizestrant versus fulvestrant was assessed in the SERENA-2 phase II trial. In contrast to the EMERALD trial, no patient received more than one line of ET; only 50% of them received CDK4/6is, and no patient received prior treatment with fulvestrant. Camizestrant also significantly prolonged PFS with 75 mg (7.2 compared to 3.7 months) and 150 mg doses (7.7 versus 3.7 months) in *ESR1*-mutated patients. Moreover, camizestrant, at both doses, decreased *ESR1*-mutant ctDNA to being not measurable or close to not measurable levels [124]. The phase III SERENA-6 trial evaluates a switch to camizestrant plus the same CDK4/6i versus the continuation of a nonsteroidal AI plus CDK4/6i (palbociclib or abemaciclib) in an HR+/HER2− advanced or relapsed breast cancer population who have measurable *ESR1* mutations in ctDNA during first-line treatment before clinical/radiological progression (NCT04964934) [125]. Imlunestrant is a SERD, which showed 6.5 months as a median PFS (3.6–8.3 months, range) when tested as a second-line treatment after ET plus CDK4/6i. The EMBER-3 study is a phase III trial, where imlunestrant alone is compared with imlunestrant in association with abemaciclib and with an investigator’s choice hormone therapy as second-line therapy after AI or AI plus a CDK4/6i. Imlunestrant alone obtained a significantly longer PFS than conventional therapy in a population with *ESR1* mutations but not in the entire sample size. Imlunestrant, in combination with abemaciclib, unlike imlunestrant alone, significantly prolonged PFS irrespective of the *ESR1* status [126]. Giredestrant (GDC-9545) is a SERD, active either as monotherapy or in association with a CDK4/6i in *ESR1* mutant or wild-type tumor models [127]. The phase II acelERA Breast Cancer Study evaluated ER+ HER2− relapsed breast cancer patients progressing after 1–2 lines of systemic therapy, including ET for at least 6 months and a targeted agent; one previous line of chemotherapy and fulvestrant were allowed; patients were randomly allocated to giredestrant or fulvestrant/AI (the physician’s choice of hormone monotherapy, PCET) until disease progression/unacceptable toxicity. After 7.9 months as the median follow-up, in the 303 recruited patients, the median PFS was 5.6 months in the giredestrant arm vs. 5.4 months in the PCET arm. At 6 months, 46.8% and 39.6% were the PFS rates in the giredestrant and PCET arms, respectively. Among the 90 *ESR1*-mutated patients, the median PFS was 5.3 months versus 3.5 months in the giredestrant and PCET arms, respectively. Despite this, the primary endpoint did not attain statistical significance, and the authors concluded that giredestrant had shown therapeutic efficacy in most subgroups and a trend toward a benefit in the population having *ESR1* mutations [128]. Trials evaluating giredestrant alone or in combination with other drugs are ongoing (NCT04802759).

### 5.2. PI3K/AKT/mTOR Pathway Inhibitors

*PIK3CA* somatic mutations occur in about 30–50% of the relapsed HR+/HER2− breast cancer population [129]. In the phase III SOLAR-1 trial, alpelisib, a PI3Kalpha-specific inhibitor, was evaluated in association with fulvestrant in patients resistant to prior ET. In patients with *PIK3CA* mutations, the combination, unlike fulvestrant alone, significantly prolonged PFS but not the OS (11.0 vs. 5.7 months, respectively) [130]. However, in this trial, only 6% of the subjects were previously given a CDK4/6i. The phase II BYLieve trial confirmed that adding alpelisib to ET is effective for relapsed HR+/HER2− breast cancer subjects carrying a *PIK3CA* mutation and progressing on CDK4/6is and ET [131]. The phase II FAKTION trial recruited women with HR+/HER2− relapsed breast cancer progressing after or during AI and with no prior CDK4/6i; patients received the AKT inhibitor capivasertib plus fulvestrant, and a significant PFS and OS benefit occurred in those carrying mutations in *PTEN*, *AKT1*, or *PIK3CA* genes [132]. The phase III CAPItello-291 trial enrolled patients progressing on AI plus CDK4/6i; subjects could have received 1–2 lines of ET and no more than one line of chemotherapy in the advanced/relapsed setting; patients who received prior treatment with fulvestrant, AKT, PI3K, and mTOR inhibitors were excluded; the enrolled subjects were or were not carriers of mutations in the genes of the PIK3CA-AKT-PTEN pathway. The capivasertib–fulvestrant association doubled the median PFS against the placebo given with fulvestrant (median PFS 7.2 vs. 3.6 months in the overall population; 7.3 months vs. 3.1 months in the AKT pathway mutated population) [133]. Inavolisib is a PIK3CA inhibitor that also induces the deterioration of the alpha isoform of the p110 catalytic subunit of PIK3CA. Inavolisib was shown to synergize with palbociclib and fulvestrant in experimental tumor models. INAVO-120 (NCT04191499) is a phase III randomized study, where the association of inavolisib with palbociclib and fulvestrant was compared with that of placebo plus palbociclib and fulvestrant as a first-line treatment in subjects with *PIK3CA*-mutated, ER+/HER2− advanced, or relapsed breast cancer who had recurred during or within one year after adjuvant ET. The median PFS was 15.0 months and 7.3 months in the inavolisib arm and in the placebo arm, respectively [134]. INAVO-121 will compare inavolisib plus fulvestrant with alpelisib plus fulvestrant in a population with *PIK3CA*-mutated, ER+/HER2− advanced, or relapsed breast cancer progressing after ET plus CDK4/6is (NCT05646862). Other AKT inhibitors, such as ipatasertib, are under evaluation in trials following the occurrence of resistance to CDK4/6is [135] (NCT04650581). The BOLERO-2 evaluated the mTOR inhibitor everolimus in patients progressing on an AI [32]. Everolimus administered in association with exemestane prolonged the median PFS (10.6 vs. 4.1 months) but not the OS [136]. Although no patient in this study had been previously given a CDK4/6i, the use of ET combined with everolimus after progression on CDK4/6is is suggested by several studies [137,138]. Dual PI3K and mTOR inhibitors are also under evaluation. Particularly, a phase III trial (VIKTORIA-1, NCT05501886) is evaluating gedatolisib, an intravenously administered pan-PI3K/mTOR inhibitor [139], in combination with fulvestrant with or without palbociclib in advanced or metastatic HR+/HER2− breast cancer that is progressing on or after AI plus CDK4/6i.

### 5.3. Antibody–Drug Conjugates (ADCs)

ADCs are composed of antibodies that target tumor antigens conjugated to a chemotherapeutic agent [140]. Upon delivery of the cytotoxic agent to breast cancer cells, the cleavable linker provokes a bystander killing, as it affects neighboring cancer cells, even if they do not express the target antigen [141]. The HER2-targeted ADC trastuzumab deruxtecan is engineered by linking a humanized anti-HER2 monoclonal antibody to a topoisomerase I inhibitor payload via a tetrapeptide cleavable linker. In the DESTINY-Breast04 phase III trial, it significantly improved outcomes in the HR+ HER2low (1+ or 2+ score by immuno-histochemistry [IHC] and negative in situ hybridization), advanced or relapsed breast cancer subjects progressing after up two different regimens of chemotherapy. The trial reported a PFS of 10 and an OS of 23.9 months compared to 5.4 and 17.5 months, respectively, with the physician’s choice of chemotherapy. Notably, patients previously treated or not with CDK4/6is experienced a comparable PFS benefit [142]. Therefore, the drug received FDA approval for this population. DESTINY-Breast-06 is a phase III trial involving ER+/HER2 low or ultralow (IHC 0 with membrane staining) metastatic breast cancer patients progressing after one or more lines of ET with no previous chemotherapy for metastatic disease. Most of these patients received prior CDK4/6is. Patients were given trastuzumab deruxtecan or one among capecitabine, paclitaxel, or nab-paclitaxel at the physician’s choice. A total of 713 of the 866 enrolled were HER2-low, and 153 were HER2-ultralow subjects. Considering the HER2 low population, the median PFS was 13.2 months in the trastuzumab deruxtecan arm and 8.1 months in the chemotherapy arm [143]; similar findings were described in HER2-ultralow patients [144]. Sacituzumab govitecan, another ADC, is an antibody against the transmembrane glycoprotein trophoblast cell-surface antigen 2 (Trop-2) linked to SN-38, a topoisomerase I inhibitor [145]. Trop-2 amplification occurs in various malignancies and in more than 90% of ER+/HER2− breast cancer cells [146]. The TROPiCS-02 phase III trial recruited ER+/HER2− breast cancer patients progressing after three different lines of systemic therapy, a CDK4/6i among them; in this setting, a significant prolonging of PFS to 5.5 months occurred with sacituzumab govitecan, versus 4 months with the physician’s choice chemotherapy; the clinical benefit was independent of Trop-2 expression. Following these results, sacituzumab govitecan received approval from the FDA for this ER+ patient cohort [147,148]. In the recent TROPION-PanTumor01 phase I trial, datopotamab deruxtecan, an ADC similar to trastuzumab deruxtecan, showed promising activity and low toxicity in heavily pretreated ER+/HER2− breast cancer subjects, most of them treated with a CDK4/6i [149]. The TROPION-Breast-01 phase III trial evaluated the clinical performance of datopotamab deruxtecan versus one among eribulin, vinorelbine, capecitabine, or gemcitabine at the investigator’s choice in advanced breast cancer subjects progressed on ET and after 1–2 prior lines of chemotherapy; those who were given datopotamab deruxtecan showed a statistically significant prolonged median PFS (6.9 vs. 4.9 months) and a good safety profile [150,151].

## 6. Other Therapeutic Strategies

### 6.1. Continuing CDK4/6is

In the MAINTAIN phase II trial, an HR+/HER2− relapsed breast cancer population progressing during ET plus CDK4/6i changed the ET (to fulvestrant or exemestane) and were randomly allocated to treatment with ribociclib or a placebo. Of the 119 recruited patients, 103 (86.5%) had been previously given palbociclib and 14 (11.7%) ribociclib. The median PFS was 5.9 months in patients allocated to the switched ET plus ribociclib arm versus 2.76 months in those allocated to the switched ET plus placebo arm. The benefit was mainly seen in the *ESR1* wild-type subset; however, a small sample size was *ESR1* mutant patients with a higher frequency of *CCND1* and/or *FGFR1* gene amplifications [118]. In contrast, in the phase II PACE trial, only a slightly prolonged PFS occurred in patients carrying *ESR1* or *PIK3CA* mutations when palbociclib with fulvestrant was continued after progression [152]. Likewise, in the PALMIRA trial, continuing palbociclib beyond progression in combination with a second-line ET did not prolong PFS in comparison with only ET as the second line [153]. Perhaps, switching to a different CDK4/6i is to be preferred rather than maintaining the same agent; different mechanisms of action and resistance among different CDK4/6is could account for the differing outcomes across these trials. The phase III post-MONARCH trial was conducted on an ER+/HER2− advanced or relapsed breast cancer population progressing on ET plus CDK4/6i as a first-line treatment. The subjects received abemaciclib + fulvestrant or placebo + fulvestrant. Nearly all enrolled subjects had been mainly treated with palbociclib and ribociclib in the advanced setting. Following a median follow-up of 13 months, the median PFS was 6.0 months in subjects receiving abemaciclib + fulvestrant and 5.3 months in those who were given fulvestrant + placebo. The PFS benefit was irrespective of *ESR1* or *PIK3CA* mutations [154].

### 6.2. Next-Generation Endocrine Agents

Complete estrogen receptor antagonists (CERANs), SERMs, and SERM/SERD Hybrids (SSHs), as well as selective estrogen receptor covalent antagonists (SERCAs), selective human ER partial agonists (ShERPAs), and PROteolysis Targeting Chimeras (PROTACs) are new anti-estrogens [155]. Selective androgen receptor modulators (SARMs) are also novel endocrine agents. Most of them are still under clinical development.

#### 6.2.1. CERANs

CERANs circumvent endocrine resistance in breast tumors by inactivating the two distinct transcriptional activation domains of the ER, such as the activation functions 1 and 2 (AF1 and AF2). AF1 is activated through various pathways like mTOR, PI3K, and MAPK, while AF2 activation occurs through the estrogen ligand itself. While SERDs and SERMs mainly target AF2, CERANs target both AF1 and AF2 [156]. Palazestrant (OP-1250) is an oral CERAN that induced a decrease in both wild-type and mutant ER breast tumors in xenograft models [157]. Currently, it is under evaluation in phase I/II trials (NCT05266105, NCT06016738).

#### 6.2.2. SSHs

SERMs act as ER antagonists by inhibiting AF2 while also showing agonist effects through AF1, depending on the cell type, and involve diverse co-activators and co-repressors. In an experimental model of AI-resistant breast cancer, lasofoxifene alone or in combination with palbociclib inhibited tumor growth more than fulvestrant, independent of *ESR1* mutation [158]. The phase II ELAINE-1 trial was conducted in HR+/HER2− relapsed breast cancer, with 40% of the subjects having the Y537S *ESR1* mutation and progressing on prior AI and CDK4/6is; lasofoxifene was given as a single agent against fulvestrant and did not significantly prolong PFS [159]. The ELAINE-2 trial examined lasofoxifene plus abemaciclib in pretreated subjects with *ESR1*-mutated metastatic breast cancer. The clinical benefit rate (CBR), ORR, and median PFS were 65.5%, 55.6%, and about 13 months, respectively [160]. The phase III ELAINE-3 study is assessing lasofoxifene + abemaciclib versus fulvestrant + abemaciclib in metastatic subjects with an *ESR1* mutation progressing during AI plus palbociclib or ribociclib (NCT05696626) [161] The SERM/SERD bazedoxifene showed clinical efficacy as an anticancer agent in HR+ endocrine-resistant breast cancer models, particularly in Y537S *ESR1* mutation carriers [162]. A phase Ib/II trial reported a CBR of 33.3% for bazedoxifene and palbociclib in these subjects, regardless of an *ESR1* mutation [163].

#### 6.2.3. SERCAs

SERCAs neutralize ER by targeting a specific cysteine residue and favoring morphological changes. In experimental models, H3B-6545 antagonized both the wild-type and mutant ER and showed greater anticancer activity than fulvestrant [164]. A phase I/II trial enrolled HER−relapsed breast cancer subjects who had already received CDK4/6is (87%), fulvestrant (71%), or chemotherapy (54%), with 58% of them carrying *ESR1* mutations. An ORR of 16.4% and a median PFS of 3.8 months occurred in the preliminary analysis with a manageable safety profile [165].

#### 6.2.4. ShERPAs

ShERPAs mimic the action of beta-estradiol binding to ER within the nucleus and prompting its extranuclear translocation, thus inhibiting the growth of ER+ tumor cells. ShERPAs showed promising efficacy in tamoxifen refractory breast cancer cells and in xenograft models [166]. TTC-352 showed antitumor activity in a phase I study in subjects progressing following two or more lines of ET, including one line with a CDK4/6i [167].

#### 6.2.5. PROTACs

PROTACs are bifunctional molecules designed to bind a particular target protein, like the ER, while concurrently recruiting an E3 ubiquitin ligase. A PROTAC, by bringing the target protein and E3 ligase together, enables the ubiquitylation of the target, which then leads to its deterioration via the ubiquitin–proteasome system. Following this, it initiates another degradation cycle [168]. Vepdegestrant (ARV-471) has been shown to be an anticancer drug that is superior to fulvestrant in xenograft models [169]. The phase I/II VERITAC trial enrolled patients who had previously received anti-estrogens and CDK 4/6is; vepdegestrant showed good clinical activity in the whole studied population and in subjects carrying *ESR1* mutations (CBR up to 38.9% and 54.5%, respectively) [170]. In pretreated HR+ HER2− patients, vepdegestrant plus palbociclib showed good activity (for CBR, 63% in ITT subjects, and 72.4% in *ESR1* mutants) [171]. The phase III VERITAC-2 trial (NCT05654623) is evaluating the clinical performance of vepdegestrant compared with fulvestrant in subjects progressing following first-line CDK4/6i and ET. Other trials are evaluating vepdegestrant in combination with various drugs, including CDK4/6is (NCT06125522, NCT06206837, NCT05548127, NCT05909397, NCT05654623, NCT05573555, and NCT04072952). AC682 is a chimeric compound that, given alone or plus CDK4/6is or PI3K/mTOR pathway inhibitors, showed anticancer activity in experimental ER+ breast cancer models, including those with *ESR1* mutations [172]. A phase I trial is currently underway (NCT05080842).

#### 6.2.6. SARMs

The androgen receptor (AR) is a steroid nuclear receptor commonly expressed in HR+HER2− breast cancer [173], and SARMs act as either AR agonists or antagonists. AR may have different roles in ER+ compared to ER-negative breast cancers. AR in HR+ breast cancer correlates with a favorable prognosis, and AR agonism inhibits the progression of both endocrine-responsive and -refractory breast cancers [174]. Enzalutamide, a nonsteroidal anti-androgen, in combination with exemestane versus exemestane alone, was tested in a randomized phase II trial but did not prolong PFS in subjects previously receiving ET [175]. Recently, it was found that the AR to ER ratio in breast cancer influences the sensitivity to AR-targeted treatments, suggesting the potential utility of enzalutamide in ER+ tumors with a low AR/ER ratio and AR agonists like RAD140 in those with a high AR/ER ratio [176]. In a phase II trial, enobosarm, a novel oral selective AR activator, achieved a CBR of 32% and 29%, depending on the dose used. Furthermore, the ORR was 48% and 0% in patients with over or less than 40% AR staining, respectively [177]. Randomized phase III trials are ongoing (NCT05065411) [178].

### 6.3. Agents Targeting CDK7 and CDK2

CDK 7 is a component of the CDK-activating kinase (CAK) complex and plays a dual role in transcription and cell-division cycle advance [179]. The anticancer efficacy of CDK7 inhibition in ER+ breast tumors is partially dependent on p53 and involves both cell cycle arrest and *c-Myc* suppression. Unlike the cytostatic effects seen with ET and CDK4/6is, CDK7 inhibition appears to exert cytotoxic effects. It also reduces ER phosphorylation at S118, although extended CDK7 inhibition can lead to increased ER signaling. Elevated c-Myc activity and intact p53 may serve as potential predictors of sensitivity to treatments based on CDK7is [180]. CDK7is showed significant anticancer activity, especially in TNBC and HR+ breast cancers [181]. The CAK complex activates p53, which is involved in DNA repair; therefore, CDK7is may be effective against aggressive luminal tumors with *TP53* alterations [181]. Samuraciclib (ICEC0942), a selective CDK7i, blocked the cell cycle and promoted apoptosis in experimental studies [182]. In the HR+/HER2− relapsed breast cancer population, progressing after an AI plus CDK4/6i, samuraciclib plus fulvestrant was responsible for a 36% CBR [111]. Samuraciclib is under evaluation, alone or in combination, in various clinical trials (NCT05963984, NCT05963997, and NCT06125522). Cyclin E/CDK2 has an important role in CDK4/6is resistance [99,104], and targeting CDK2 either with or without responsiveness to CDK4/6is is under evaluation. In preclinical models, dinaciclib, a non-selective CDK2i, combined with palbociclib and letrozole, was more efficacious than palbociclib and/or letrozole only [94]. Other CDK2is are under investigation in phase I/II trials in ER+/HER2− breast cancer patients (NCT04553133 and NCT05252416). In the VELA phase I trial (NCT05252416), the CDK2i BLU-222, administered alone in HR+/HER2− breast and further different advanced solid cancers, showed clinical efficacy and low toxicity [183]. Likewise, in another phase I/II trial (NCT04553133), the novel CDK2i PF-07104091 showed clinical efficacy in a previously largely treated advanced/relapsed breast cancer population progressing during ET plus CDK4/6is [184]. Similarly, an ongoing phase 1/2 trial (NCT03519178) is evaluating CDK2/CDK4/CDK6i PF-06873600 alone or with letrozole or fulvestrant in the same population progressing after CDK4/6is, ET and ≤2 lines of chemotherapy.

### 6.4. Immune Therapy (IT)

IT includes ESRmut vaccines, a beta-interferon-interleukin-2 sequence given in association with ET, or treatment with macrophage inhibitors. Mutations in the *ESR1* gene can account for neo-epitopes that could be targeted through IT. The in vitro cytotoxicity assays demonstrated that expanded antigen-specific CTLs could effectively lyse peptide-pulsed targets and breast cancer cells expressing five peptides as the most immunogenic candidates, derived from D538G, Y537S, and E380Q, which are the three most prevalent ESR1 mutations [185]. Accordingly, in a study using high-performance liquid chromatography (HPLC), the presentation of ESR1 and ESR1mut peptides on human MHC was shown in an ER+ breast cancer subject along with the presence of human T cells that were reactive to ESR1mut epitopes [186]. This observation supported the development of ESR1mut vaccines. Since the early nineties, our research group discovered a synergism of a beta-interferon-interleukin-2 sequence when administered in association with conventional ET in an ER+ endocrine-dependent relapsed breast cancer population. Since then, a few patients were recruited in a pilot study where they received this IT first combined with tamoxifen, which, at that time, was the first-line standard ET, and, thereafter, with AIs in some others. The initial findings were reported several times [187,188,189], and recently [190], the final results were discussed with 95 controls and 42 cases overall being retrospectively compared. The 95 controls were ER+/HER2− relapsed breast cancer subjects who had undergone first-line ET with AIs or fulvestrant. Twenty-eight of them (28.9%) were also given biological drugs, CDK4/6is being among them. The 42 cases were ER+ endocrine-dependent relapsed subjects who were given a beta-interferon-interleukin-2 sequence plus first-line ET. In particular, 39 (92.9%) subjects received SERMs/SERDs, and the three remaining AIs were administered. The median PFS and OS were significantly prolonged in the 42 studied patients compared with the 95 controls (median time 33 vs. 18 months, *p* = 0.002, and 81 vs. 62 months, *p* = 0.019, respectively) (Figure 3). We hypothesized that “in responsive metastatic disease a stable or decreased tumor burden and a lower genetic instability due to the quiescent state (G0-G1 state) of tumor cells induced by anti-estrogens is also likely to reduce the immune evasion and immune inhibition. This favors the immune attack stimulated by the concomitant immune therapy” [188]. This mechanistic rationale is widely different from that of CDK4/6is; therefore, the proposed association can be an alternative to CDK4/6is or a reasonable choice for patients who maintain ER signaling after they have progressed with ET and CDK4/6is. Table 1 summarizes the main characteristics of the clinical studies carried out with current and experimental therapies in endocrine-CDK4/6is-resistant advanced ER+/HER2− breast cancer.

### 6.5. Further Drugs and Targets

These include FGFR inhibitors, inhibition of MITF or MYSM1, quercetin, and agents targeting the CXCR1/2 receptor.

The FGFR inhibitors erdafitinib, lucitanib, and dovitinib have reported conflicting results, reflecting the complication of FGFR signaling in breast cancer. Thus, in the next studies, defining biomarkers and combined treatments to improve the effectiveness of FGFR-targeted therapies should be prioritized [38]. Inhibition of MITF or its O-GlcNAcylation restores sensitivity to palbociclib in resistant cells. Therefore, targeting MITF is a potential strategy for managing CDK4/6i-resistant breast cancer [93]. MYSM1, a deubiquitinase, is an epigenetic regulator of ERα activity and is relevant in tumor growth. Silencing MYSM1 inhibits the increase of breast cancer-derived cells in xenograft models and increases their sensitivity to anti-estrogens. Virtual screening identified Imatinib as a small molecule capable of interacting with the catalytic MPN domain of MYSM1, and this effectively inhibited breast cancer cell growth, highlighting the MYSM1-ERα axis as a promising target to overcome endocrine independence in breast cancer [19]. In silico analyses likely indicate quercetin as a CDK2 inhibitor in ER+ breast cancer [191] and that quercetin can re-sensitize breast cancer cells to palbociclib by governing the circHIAT1/miR-19a-3p/CADM2 axis. Therefore, quercetin can revert lack of sensitivity to palbociclib and can hinder EMT [91]. Endocrine-resistant breast cancer (ERBC) cells secrete CXCL1, which links with the CXCR1/2 on fibroblasts, activating ERK/MAPK signaling and inducing CXCL1 amplification. CXCL1, in turn, interacts with CXCR1/2 on ERBC cells, further activating the ERK/MAPK pathway and promoting ERBC cell proliferation and spread, thus highlighting CXCL1 as a key element in ERBC proliferation. Accordingly, reparixin in association with CDK4/6is, is promising to counteract endocrine-refractory breast cancer progression and diffusion [192].

## 7. Discussion and Conclusions

In the last decade, the survival of the recurred ER+ breast cancer population has improved, and CDK4/6is’s introduction into current clinical practice, in addition to ET, was a main novel therapeutic strategy. However, inevitably, resistance develops over time, and a better comprehension of the reasons accounting for resistance to anti-estrogens and/or CDK4/6is has paved the way for new drugs and treatment modalities. In patients maintaining ER-dependent signaling, fulvestrant, a SERD, as a single agent after progression on CDK4/6is, demonstrated limited efficacy in the phase III trials. Better results have been reported with fulvestrant given as second-line ET in addition to CDK4/6is, alpelisib, and everolimus. Namely, alpelisib, a PI3K inhibitor, plus fulvestrant, has been approved for ER+/HER2−, *PIK3CA*-mutated advanced breast cancer patients following prior ET. Capivasertib, an AKT inhibitor, plus fulvestrant, has shown important improvement in PFS and OS in HR+/HER2− relapsed breast cancer subjects with mutated *PTEN*, *AKT1*, or *PIK3CA* genes and progressing after AI plus CDK4/6i. Other AKT inhibitors are under evaluation after progression during CDK4/6is. Everolimus, an mTOR inhibitor, is currently used in addition to exemestane in HR+/HER2− relapsed breast cancer populations progressing after ET, with or without CDK4/6is. Meanwhile, other SERDs have been investigated as elacestrant, camizestrant, and imlunestrant, which are currently under evaluation alone or in combination with different biological drugs. Among the more recent SERMs, lasofoxifene in association with abemaciclib is under evaluation in relapsed subjects with *ESR1* mutations progressing on an AI plus CDK4/6i. Vepdegestrant, a PROTAC, is currently being assessed compared with fulvestrant in subjects progressing following first-line therapy with a CDK4/6i and ET. Continuing CDK4/6is therapy after progression has shown contrasting results, while replacing it with another CDK4/6i may be more effective. However, the phase III PADA-1 study, conducted in patients submitted to first-line therapy with a CDK4/6i and an AI, showed an improvement in PFS in patients who replaced AI with fulvestrant when early *ESR1* mutation detection before radiological progression and who were continuing CDK4/6i therapy occurred. Samuraciclib, a CDK7i, given in addition to fulvestrant, showed efficacy in subjects with HR+/HER2− relapsed breast cancer progressing during an AI plus a CDK4/6i. Several CDK2is, alone or in combination with ET, are under investigation in phase I/II clinical trials in ER+/HER2− breast cancer patients. As for immunotherapy, ER+ breast cancer is commonly thought to be a “cold” tumor, and checkpoint inhibitors did not show significant activity. Despite this, since the early nineties, our research group discovered a synergism of beta-interferon-interleukin-2 sequence given in association with conventional ET in the ER+ relapsed breast cancer population. The proposed mechanistic rationale of this IT is widely different from that of CDK4/6is; therefore, its association with ET can be a reasonable first choice or an alternative for patients progressing after ET and CDK4/6is. In conclusion, overcoming resistance to ET is a “work in progress”, and future studies are expected to better select patients for different therapeutic strategies based on more specific biological and/or genetic markers. Liquid biopsy may provide a real-time portrait of the disease state with insights into disease biology, including the processes responsible for ET independence and cancer proliferation. A few different liquid-based biomarkers with prognostic and predictive capabilities in ER+/HER2− breast cancer are going to be available in the current clinical practice [193,194]. Next-generation sequencing (NGS) is an important diagnostic tool that can provide a comprehensive analysis of PI3K pathway alterations and other actionable markers, such as *ESR1* and *BRCA*. Due to this, NGS of tumor or plasma samples should become the standard of care for patients with HR+/HER2− MBC [195].

## Figures and Tables

**Figure 1 ijms-26-03438-f001:**
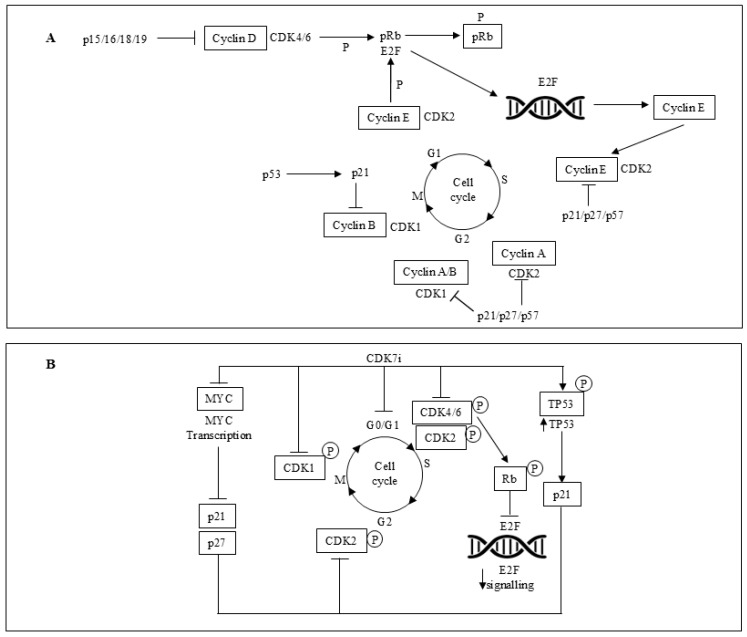
(**A**) Complexity of the molecular pathways governing the function of the cyclin-dependent kinases (CDKs) through the cell-division cycle. (**B**) Key mechanisms of CDK7 and CDK7 inhibition in ER+ breast cancer cells. The cell-division cycle starts depending on the tangled interplay of CDK4 and CDK6 complexes, which involve D-type cyclins and the release of E2F transcription factors by pRb phosphorylation. They regulate the transcription of genes that play a key role in S-phase advance. S-phase progression is then regulated by the cooperation of CDK2-cyclin E/cyclin A complexes. CDK function arrangement is promoted by CDKis, particularly in addressing CDK4/6 or CDK2. Progression from the G2 phase to the M phase is governed by the collaborating A and B cyclins along with CDK1. CDK: cyclin-dependent kinase; p: protein; Rb: retinoblastoma; P: phosphorylation; CDK7i: cyclin-dependent kinase 7 inhibition; E2F: early region 2 binding factor; *TP53*: tumor protein p53. Also, see the text.

**Figure 2 ijms-26-03438-f002:**
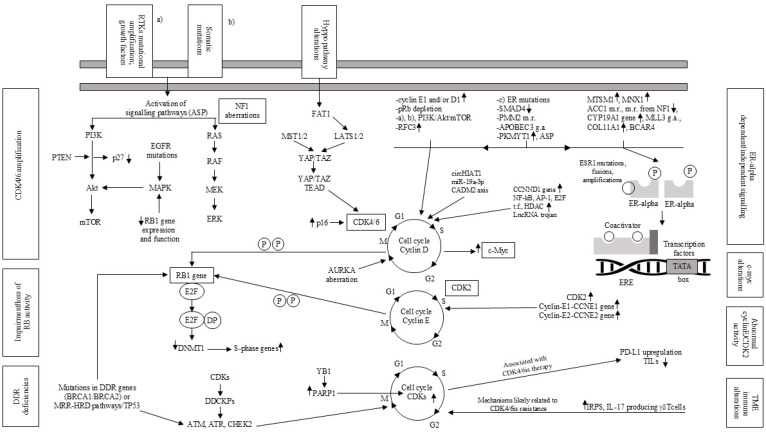
The main mechanisms of resistance to ET and/or CDK4/6is in advanced ER+/HER2− breast cancer. (a) These include ERBB2, FGFR1/2, EGFR, IGFR1, and NOTCH; (b) in Akt1, PI3KCA, PTEN, HRAS, KRAS, and NRAS; (c) ESR1 Y537S/Y537N mutations; RTK: receptor tyrosine kinase; PI3K: phosphoinositide 3 kinase; PTEN: phosphatase and tensin homolog; Akt: protein kinase B; mTOR: mammalian target of rapamycin; EGFR: epidermal growth factor receptor; ERBB2: receptor tyrosine-protein kinase erbB2; FGFR1/2: fibriblast growth factor receptor 1/2; IGFR1: insulin-like growth factor receptor 1; NOTCH: notch signaling pathway; MAPK: mitogen-activated protein kinase; Rb: retinoblastoma; RAS: rat sarcoma virus protein superfamily; *NF1*: neurofibromatosis type 1 gene; RAF: rapidly accelerated fibrosarcoma protein kinase; MEK: mitogen-activated protein kinase kinase; ERK: extracellular signal-regulated kinase; FAT1: protein encoded by *FAT* gene that takes part of the cadherin superfamily proteins; MST1/2: macrophage-stimulating protein 1/2; LATS1/2: large tumor suppressor kinase 1/2; YAP: yes-associated protein-1; TAZ: transcriptional adaptor putative zinc finger; TEAD: TEA domain family member; CDK: cyclin-dependent kinase; p: protein; SMAD4: smad family member 4; PMM2 m.r.; phosphomannomutase 2 metabolic reprogramming; APOBEC3 g.a.: apolipoprotein B mRNA-editing enzyme, catalytic subunit 3, genetic alteration; PKMYT: membrane-associated tyrosine and threonine specific cdc 2-inhibitory kinase; MYSM1: Myblike, SWIRM and MNP domain 1; MNX1: motor neuron and pancreas homeobox 1; ACC1 m.r.: acetyl-CoA carboxylase 1 metabolic reprogramming; CYP19A1: member of the cytochrome P450 superfamily; MLL3 g.a.: myeloid lymphoid or mixed lineage leukemia protein 3 genetic alterations; COLL11A1: collagen, type XI, alpha 1; BCAR4: breast cancer anti-estrogen resistance protein 4; RFC3: replication factor C subunit 3; AEs: anti-estrogens; ERalpha: estrogen receptor alpha; ERE: estrogen receptor element; TATA: tiamine-adenine sequence; *c-myc*: family of genes overexpressed in various cancers and homologous with an avian virus; circHIAT1: circular RNA HIAT1; miR-19a-3p: microRNA 19a-3p; CADM2: cell adhesion molecule 2; NF-κB: nuclear factor kappa-light chain enhancer of activated B cells; AP-1: activating protein-1; E2F: transcription factor E2F; t.f.: franscription factor; HDAC: hystone deacetylase; LncRNA: long non-coding RNA; AURKA: aurora kinase A; *CCNE1/2*: genes encoding cyclin E1/E2 proteins; DNMT1: DNA methyltransferase 1; DDR: DNA damage repair; *BRCA1/2*: breast cancer type 1/2 genes; HRR: homologous recombination repair; HRD: homologous repair deficiency; *TP53*: tumor protein 53; DDCKPS: DNA damage checkpoints signaling; ATM: part of serine/threonine protein kinase ATM; ATR: ataxia-teleangectasia and RAD3 related; CHEK2: checkpoint kinase 2; PD-L1: programmed death ligand 1; TILs: tumor-infiltrating lymphocytes; IRPS: interferon-related palbociclib-resistance signature; IL: interleukin; P: phosphorylation. increase. Decrease. (Also see text).

**Figure 3 ijms-26-03438-f003:**
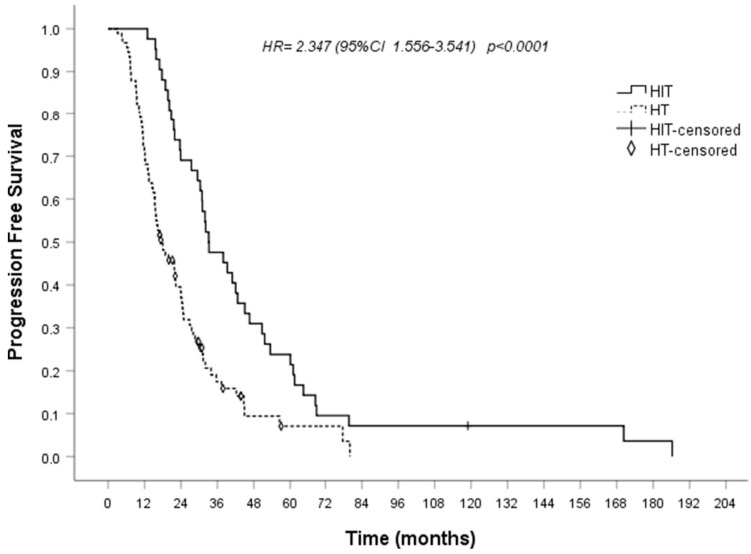
Progression-free survival correlated to hormonal therapy (survival median time of HIT 33.1 (95% CI 24.5–41.8); survival median time of HT 18 (95% CI 12.1–23.8)). HIT: hormone immunotherapy; HT: hormone therapy. Reproduced with permission by Nicolini A., et al. J. Cancer Metastasis Treat. 2022 8, 13 [190].

**Table 1 ijms-26-03438-t001:** Current and experimental therapies in advanced endocrine-CDK4/6is-resistant ER+/HER2− breast cancer. (**A**) Fulvestrant and novel oral SERDs given alone or in association with other drugs; (**B**) PI3K/Akt/mTOR inhibitors plus fulvestrant or exemestane; (**C**) antibody-drug conjugates; (**D**) continuing the same or other ET along with the same or a different CDK4/6i; (**E**) next-generation endocrine agents (CERANs, SSHs, PROTACs, and SARMs); CDK7 inhibitors and a novel schedule of hormone immunotherapy.

(**A**)				
**Study (Phase)**	**Therapy**	**Setting**	**Main Endpoints**	**Reference**
CONFIRM (III)	Fulvestrant 500 mg vs. 250 mg	Postmenopausal MBC women progressing on ET	mPFS 6.5 months mOS 26.4 months	[115]
MAINTAIN (Control arm)	Fulvestrant 500 mg	ABC patients progressed on ET plus a CDK4/6i	mPFS 2.76 months	[118]
VERONICA (Control arm)	Fulvestrant 500 mg	MBC patients, post CDK4/6i progression	m PFS 1.94 months	[119]
EMERALD (III)	Elacestrant vs. standard ET	ABC patients after 1–2 lines of ET, including a CDK4/6i and ≤1 chemotherapy	6-month PFS rate: 34.3% vs. 20.4% in all patients; 40.8% vs. 19.1% in patients with *ESR1* mutation 12-month PFS rate: 22.3% vs. 9.4% in all patients; 26.8% and 8.2% in patients with *ESR1* mutation	[121]
ELEVATE (Ib/II)	Elacestrant in combination with alpelisib, or capivasertib, or everolimus, or palbociclib, or abemaciclib, or ribociclib	ABC/MBC, progressing on one or up to two prior lines of ET (inclusion criteria differ in every study arm)	Primary: PFS, safety Secondary: ORR, DOR, CBR, OS	[123]
ADELA (III)	Elacestrant plus everolimus vs. elacestrant alone	ABC patients with *ESR1*-mutated tumors progressing on ET plus CDK4/6i	Primary: PFS Secondary: OS, ORR, CBR, DOR, safety and quality of life	NCT06382948
SERENA-2 (II)	Camizestrant once daily at 75 mg, 150 mg, or 300 mg vs. fulvestrant at 500 mg	ABC patients progressing on at least one line of ET and no more than one previous ET in the advanced setting	mPFS 7.2 months with camizestrant 75 mg, 7.7 with 150 mg, vs. 3.7 months with fulvestrant	[124]
SERENA-6 (III)	Camizestrant plus CDK4/6i maintaining	ABC patients progressing on AI plus CDK4/6i upon detection of *ESR1*m in ctDNA before clinical disease progression	Primary endpoint: PFS; secondary: CT-free survival, PFS2, OS, safety	NCT04964934
EMBER-3 (III)	Imlunestrant vs. monoET (88% fulvestrant) vs. imlunestrant plus abemaciclib	ABC patients recurring or progressing during or after AI alone or AI plus a CDK4/6i (59.8%)	Patients with *ESR1* mutations: mPFS 5.5 months with imlunestrant vs. 3.8 with monoET Overall population: mPFS 5.6 months with imlunestrant vs. 5.5 with monoET mPFS 9.4 months with imlunestrant plus abemaciclib vs. 5.5 with imlunestrant regardless of *ESR1*-mutation status	[126]
acelERA Breast Cancer Study (II)	Giredestrant vs. physician’s choice of endocrine monotherapy (PCET)	MBC patients progressing after 1–2 lines of systemic therapy; 1 previous targeted agent, 1 line of chemotherapy, and prior fulvestrant were allowed	Overall population: mPFS 5.6 months in the giredestrant arm vs. 5.4 months in the PCET arm. *ESR1* mutated patients: mPFS 5.3 months vs. 3.5 months.	[128]
MORPHEUS-Breast cancer (I-IIb umbrella study) Cohort 1	Giredestrant alone vs. giredestrant in combination with abemaciclib, or ipatasertib, or inavolisib, or ribociclib, or everolimus, or samuraciclib, or atezolizumab	ABC/MBC patients progressing on AI + CDK4/6i as first or second line	Primary: ORR, safety Secondary: PFS, DCR, CBR, OS, DOR	NCT04802759
(**B**)				
**Study (Phase)**	**Therapy**	**Setting**	**Main Endpoints**	**Reference**
SOLAR-1 (III)	Alpelisib plus fulvestrant vs. placebo plus fulvestrant	ABC patients progressing on or after AI	mOS 39.3 vs. 31.4 months in the PIK3CA-mutated cohort	[130]
BYLieve (II)	Alpelisib plus fulvestrant	ABC patients with tumor *PIK3CA* mutation, progressing on or after previous ET, including CDK4/6i	53.8% of patients were alive without disease progression at 6 months, after a median follow-up of 21.8 months	[131]
CAPItello-291 (III)	Capivasertib plus fulvestrant vs. placebo plus fulvestrant	ABC patients relapsed or progressed during or after treatment with an AI, with or without CDK4/6i	mPFS 7.2 vs. 3.6 months in the overall population mPFS 7.3 vs. 3.1 months in patients with AKT pathway alterations	[133]
INAVO-120 (III)	Inavolisib plus palbociclib-fulvestrant vs. placebo plus palbociclib-fulvestrant	*PIK3CA*-mutated ABC/MBC patients relapsed during or within 12 months after the completion of adjuvant ET	mPFS 15.0 months vs. 7.3 months ORR 58.4% vs. 25.0%	[134]
INAVO-121 (III)	Inavolisib plus fulvestrant vs. alpelisib plus fulvestrant	*PIK3CA*-mutated ABC/MBC patients progressing after ET plus CDK4/6i	Primary: PFS Secondary: ORR, CBR, DOR, OS	NCT05646862
BOLERO-2 (III)	Everolimus plus exemestane vs. placebo plus exemestane	Patients who had recurrence or progression on NSAI	mPFS 6.9 months vs. 2.8 months	[32]
Retrospective	Everolimus plus ET	MBC patients progressing on palbociclib	mPFS 4.2 months, mOS 18.7 months	[137]
Retrospective	Everolimus plus exemestane	MBC progressing on an NSAI alone or in association with a CDK4/6i (40%)	No significant difference in mPFS (3.6 vs. 4.2 months) or mOS (15.6 vs. 11.3 months) between patients who had received prior CDK4/6is and those who had not	[138]
IPATunity-150 (Ib)	Ipatasertib plus fulvestrant plus palbociclib	Patients relapsed during adjuvant ET who had not previously received a CDK4/6i	ORR 45%, mDOR 9.6 months	[135]
FINER (III)	Ipatasertib plus fulvestrant vs. placebo plus fulvestrant	ABC patients progressing on AI plus CDK4/6i	Primary: PFS Secondary: PFS in *PIK3CA/AKT1/PTEN* altered and non-altered cohorts, RR, DOR, CBR, OS, TSST, safety	NCT04650581
VIKTORIA-1 (III)	Gedatolisib plus fulvestrant with or without palbociclib vs. standard-of-care ET	ABC/MBC patients progressing on AI plus CDK4/6i	Primary: PFS Secondary: OS, ORR, DOR, TTR, CBR	NCT05501886
(**C**)				
**Study (Phase)**	**Therapy**	**Setting**	**Main Endpoints**	**Reference**
Destiny-Breast-04 (III)	TDX-d vs. physician’s choice of CT.	HR+/HER2low and HR-/HER2low MBC patients progressing after CT for metastatic disease or within 6 months after completing adjuvant CT; HR+ patients must have received at least 1 line of ET	mPFS in HR+ cohort 10.1 vs. 5.4 months	[142]
Destiny-Breast-06 (III)	TDX-d vs. physician’s choice of CT	ER+/HER2 low or ultralow MBC patients progressing after one or more lines of ET and no previous CT for MBC	HER2-low disease: mPFS 13.2 months vs. 8.1 HER2-ultralow disease: mPFS 13.2 months vs. 8.3	[143,144]
TROPiCS-02 (III)	Sacituzumab-Govitecan vs. physician’s CT	MBC patients progressed after at least two prior systemic CT regimens for metastatic disease. Patients must have received at least one taxane, at least one ET, and at least one CDK4/6i	mPFS 5.5 vs. 4.0 months	[147,148]
TROPION-Breast-01 (III)	Dato-DXd vs. ICCT (eribulin or vinorelbine or capecitabine or gemcitabine)	ABC patients progressing on ET and after having received 1–2 prior lines of CT	mPFS 6.9 vs. 4.9 months At 12 months, 25.5% of patients in the Dato-DXd arm versus 14.6% in the ICCT arm were progression-free OS data immature	[150,151]
(**D**)				
**Study (Phase)**	**Therapy**	**Setting**	**Main Endpoints**	**Reference**
MAINTAIN (II)	Stop CDK4/6i, switch ET (fulvestrant or exemestane), and randomize to receive ribociclib or placebo	MBC patients progressing during ET and CDK4/6i (86.5% palbociclib and 11.7% ribociclib)	mPFS 5.29 months in patients assigned to switched ET plus ribociclib, 2.76 in patients switched ET plus placebo	[118]
PACE (II)	Fulvestrant (F) vs. fulvestrant plus palbociclib (F + P) vs. fulvestrant plus palbociclib and avelumab (F + P + A)	MBC patients progressing on previous AI plus CDK4/6i (90.9% palbociclib)	mPFS: 4.8 months on F, 4.6 months on F + P, 8.1 on F + P + A The difference in PFS with F + P and F + P + A versus F was greater among patients with baseline *ESR1* and *PIK3CA* alteration	[152]
PALMIRA (II)	Palbociclib (P) maintenance plus second-line ET	ABC patients progressing on first-line palbociclib plus ET (AI or fulvestrant)	PFS was 4.2 months (95% CI 3.5–5.8) in the P + ET vs. 3.6 months (95% CI 2.7–4.2) in the ET arm (hazard ratio 0.8, 95% CI 0.6–1.1, *p* = 0.206). The 6-month PFS rate was 40.9% and 28.6% for P + ET and ET, respectively. Among 138 pts with measurable disease, no significant differences were observed in ORR (6.4% vs. 2.3%) or CBR (33.0% vs. 29.5%) for P + ET and ET, respectively	[153]
Post-MONARCH (III)	Abemaciclib + fulvestrant vs. placebo + fulvestrant	MBC patients progressing on prior CDK4/6i plus AI or recurred on/after adjuvant CDK4/6i + ET	mPFS 6.0 versus 5.3 months, regardless of *ESR1* or *PIK3CA* mutations	[154]
(**E**)				
**Study (Phase)**	**Therapy**	**Setting**	**Main Endpoints**	**Reference**
Opera-01 (III)	Palazestrant vs. standard ET (fulvestrant or an AI)	ABC patients progressing after 1 or 2 prior lines of standard ET, including a CDK 4/6i	Primary: dose selection, PFS Secondary: OS, ORR, CBR, DOR, safety in patients with and without *ESR1* mutation	NCT06016738
ELAINE-2 (II)	Lasofoxifene plus abemaciclib	*ESR1*-mutated MBC patients progressing on prior ET, including CDK4/6i	mPFS 56.0 weeks PFS rates at 6, 12, and 18 months: 76.1%, 56.1%, and 38.8%, respectively. CBR at 24 weeks: 65.5%	[160]
ELAINE-3	Lasofoxifene plus abemaciclib vs. fulvestrant plus abemaciclib	ABC patients with ≥1 acquired *ESR1* mutation, progressing on AI plus palbociclib or ribociclib; ≤1 line of chemotherapy in the advanced/metastatic setting	Primary: PFS Secondary: ORR, OS, CBR, DOR, TTR, safety	NCT05696626
VERITAC (I–II)	Vepdegestrant	MBC patients pretreated with anti-estrogens plus CDK 4/6is	CBR up to 38.9% and 54.5% in the overall population and in patients with *ESR1*-mutated tumors, respectively	[170]
I−II (NCT04072952)	Vepdegestrant plus palbociclib	Pretreated MBC patients	CBR 63% in ITT population, 72.4% in *ESR1* mutant patients	[171]
VERITAC-2 (III)	Vepdegestrant vs. fulvestrant	Patients progressing after 1st line ET plus CDK4/6i	Primary: PFS in the ITT population and *ESR1* mutation-positive subpopulation. Secondary: OS, RR, safety	NCT05654623
II (NCT02463032)	Enobosarm 9 mg or 18 mg daily	Pretreated MBC patients	CBR of 32% and 29%, depending on the dose used. ORR 48% and 0% in patients with more or less than 40% AR staining, respectively	177
ENABLAR-2 (2-staged, phase III)	Enobosarm +/− Abemaciclib	AR+ER+HER2− MBC patients progressing after ET + Palbociclib or Ribociclib	PFS, OS, ORR, CBR	[178]
Multimodular I–II (NCT03363893)	Samuraciclib (dose escalation) plus fulvestrant	MBC patients progressing after an AI plus a CDK4/6i	CBR 36%	[111]
SUMIT-BS (II)	Samuraciclib plus fulvestrant versus fulvestrant alone	ABC/MBC patients progressing after AI plus CDK4/6i	Primary: CBR Secondary: ORR, DOR, PFS, safety	NCT05963984
SUMIT-ELA (Ib-II)	Samuraciclib plus elacestrant	ABC/MBC patients progressing after AI plus CDK4/6i	Primary: dose, PFS Secondary: ORR, CBR, DOR, safety	NCT05963997
Retrospective observational study	ET in combination with cyclic interferon beta and interleukin-2	Endocrine-dependent MBC patients (also see text)	mPFS 33 months mOS 81 months	[190]

ER+: estrogen receptor-positive; HER2−: epidermal growth factor receptor 2 negative; MBC: metastatic breast cancer; ABC: advanced breast cancer; ET: endocrine therapy; AI: aromatase inhibitor; NSAI: nonsteroidal aromatase inhibitor; CDK4/6i: cyclin-dependent kinases 4 and 6 inhibitor; ctDNA: circulating tumor DNA; CT: chemotherapy; *ESR1*: ESR1 gene which encodes ER-alpha; PIK3CA: phosphatidylinositol-4,5-bisphosphate 3-kinase catalytic subunit alpha; Akt: protein kinase B; PTEN: phosphatase and tensin homolog; TDX-d: trastuzumab deruxtecan; Dato-DXd: Datopotamab deruxtecan; AR: androgen receptor; m: median; PFS: progression-free survival; OS: overall survival; ORR: overall response rate; DOR: duration of response; CBR: clinical benefit rate; DCR: disease control rate; TTR: time to response; PFS2: time to second progression event; TSST: time to commencement of subsequent line of systemic therapy or death; IC: investigator’s choice; ITT: intention-to-treat.

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
