# Peer review of "Molecular Mechanisms and Therapeutic Strategies to Overcome Resistance to Endocrine Therapy and CDK4/6 Inhibitors in Advanced ER+/HER2− Breast Cancer"

_ijms, 2025, doi:10.3390/ijms26073438_

Round 1

Reviewer 1 Report

Comments and Suggestions for Authors

Recommendation: This article could be acceptable, but after a major revisions as followings:

Comments:

In this manuscript, the authors considered that in this review article, which the title is:”Molecular Mechanisms and Therapeutic Strategies to Over- 2 come Resistance to Endocrine Therapy and CDK4/6 Inhibitors 3 in Advanced ER+/HER2- Breast Cancer ”,

They have achieved: “in most cases of ER+ advanced disease, endocrine therapy (ET) is the initial treatment with different drugs that act by inhibiting the ER signaling, mainly tamoxifen, a selective estrogen receptor modulator (SERM), or fulvestrant, a selective estrogen receptor 15 degrader (SERD), or by impeding the estrogen formation as aromatase inhibitors (AIs).” However, hormone resistance intrinsic or acquired, eventually develops, making disease progression unavoidable.and their goal focuses on the main mechanisms of resistance to ET, whether used alone or in combination with biological agents,and on new drugs/strategies currently in use or under investigation to overcome it.

The topic:“Molecular Mechanisms and Therapeutic Strategies to Overcome Resistance to Endocrine Therapy and CDK4/6 Inhibitors 3 in Advanced ER+/HER2- Breast Cancer “ is very interesting and should be very important for organic and pharmaceutical chemists. This article could be acceptable after a major revisions as followings:.

  • Thepart of molecular mechanisms and therapeutic strategies to overcome resistance to endocrine therapy and CDK4/6 Inhibitors 3 in advanced ER+/HER2- breast cancer have been reviewed, but the mechanism schemes should also provided, Please draw them by yourselves, seen in the following attached model:

2) The logic for desribing the promising strategy for breast cancer therapy, or how the authors arrange strategy, should also be given and summarized the specific details .

3) The indexed literature is too many, please delete some, if possible!

Mei Luo

HFUT

Hefei, anhui, china, 230009

Comments on the Quality of English Language

The Extensive editing of English language and style required.

Author Response

Reviewer 1

Question

The English could be improved to more clearly express the research.

Answer

A professional english language editing of the entire manuscript has been carried out.

Question

The part of molecular mechanisms and therapeutic strategies to overcome ………...   have been reviewed, but the mechanism schemes should also provided, Please draw them by yourselves, seen in the following attached model:

Answer

The main mechanisms of resistance to endocrine therapy and/or CDK4/6is in advanced ER+HER2- breast cancer have been schematically grouped and shown in Figure 2. They, according to the the text, are multiple and in some instances the same agent (pathway or gene or protein) is involved in different of them. This accounts for the complexity of the illustration where the main mechanisms have been summarized at the best. However, the reader can get help by the text and the related references as suggested in the legenda of this illustration. On the other hand, too many illustrations should have been necessary to parcel out one or two mechanisms from all the others.

Question

The logic for describing the promising strategy for breast cancer therapy, or how the authors arrange strategy, should also be given and summarized the specific details

Answer

Consistent with the current scientific literature, the promising strategies for breast cancer therapy have been reported in the different subsections of the main section entitled “Common therapeutic strategies to overcome resistance to ET and/or CDK4/6is”. Moreover, in the immunotherapy subsection of the section entitled “Other therapeutic strategies”, our contribution  to the field  is also included.

Question

The indexed literature is too many, please delete some, if possible!

Answer

All the references where it was possible were removed; thus 12 references have been deleted.

Reviewer 2 Report

Comments and Suggestions for Authors

The manuscript provides a comprehensive review of resistance mechanisms to endocrine therapy (ET) and CDK4/6 inhibitors in ER+/HER2- breast cancer. It discusses molecular alterations, signaling pathways, and therapeutic strategies to overcome resistance. The article is well-structured and covers relevant aspects related to endocrine therapy resistance. However, a few corrections are recommended to enhance clarity and overall readability.

  • Consider including CDK7 in Figure 1, as it is discussed later in the manuscript and plays a significant role in both cell cycle regulation and transcriptional control. Its inclusion would offer a more complete representation of upstream regulation of the CDK network. Additionally, the diagram could benefit from further emphasis on the specific roles of key CDKs in the regulation of the cell cycle—for example, indicate the influence of CDK4/6 in G1 phase progression—this would better illustrate the sequential control of the cell cycle.

  • In Table 1, I recommend adding subsection headings to better organize the therapeutic strategies presented. For instance, Section A refers to CDK4/6 inhibitors with Fulvestrant and novel SERDs. Adding an appropriate heading to the groupings would add clarity for the reader. Furthermore, please consider referencing the relevant parts of the manuscript where these tables apply.

  • Some parts of the manuscript require rephrasing:

Page 2, Line 43: The addition of cyclin-dependent kinase 4 and 6 inhibitors (CDK4/6is) to endocrine therapy (ET) has recently been associated with improved progression-free survival (PFS) and, in some cases, overall survival (OS) in patients with invasive ER+ breast cancer. By blocking the cell cycle in the G1 phase and halting DNA synthesis, CDK4/6is act synergistically with ET.

Page 2, Line 69: However, the link between ESR1 amplifications and ET resistance, particularly reduced sensitivity to tamoxifen, remains a subject of ongoing debate

Page 2, Line 87: A brief introduction or background on SMAD4 will help understand its relevance in the context of the provided information.

Page 4, line 149: Some studies have identified a link between metabolic reprogramming and response to endocrine therapy.

Page 4, line 155: In particular, reducing PMM2 levels re-sensitized

Page 4, line 158: In ER+ breast cancer cells maintained long-term without estrogens (LTED cells), a model commonly used to study resistance to aromatase inhibitors, an increase in intracellular lipid accumulation was found.

Page 4, line 183, line 186: Please change removal to knockdown

Page 10, line 117: please use upregulate CDK6 instead of increase

Page 11, line 161: A study [110] found that PD-L1 levels are regulated by cyclin D–CDK4 and the cullin 3–SPOP E3 ligase through proteasome-mediated degradation

Page 11, line 176: Parental and estrogen deprivation–refractory MCF7 and T47D cells were used to generate palbociclib-resistant breast cancer sublines (PalboR).

Page 12, line 203: While cells treated with the CDK2i PF3600 [115] may continue to proliferate by utilizing  CDK1, combining CDK2 and CDK4/6 inhibitors prevents this compensatory mechanism.

Page 29, line 2: These include FGFR inhibitors, inhibition of MITF or MYSM1, quercetin, and agents targeting the CXCR1/2 receptor.

Author Response

Reviewer 2

Question

Consider including CDK7 in Figure 1, as it is discussed later in the manuscript and plays a significant role in both cell cycle regulation and transcriptional control. Its inclusion would offer a more complete representation of upstream regulation of the CDK network. Additionally, the diagram could benefit from further emphasis on the specific roles of key CDKs in the regulation of the cell cycle—for example, indicate the influence of CDK4/6 in G1 phase progression—this would better illustrate the sequential control of the cell cycle.

Answer

Figure 1 has been divided in two parts, part A (above) that comprehends the entire initial illustration yet reduced in size; moreover, as suggested, part B (below) has been added; this part B schematically shows the significant role that CDK7/CDK7is play in both cell cycle regulation and transcriptional control. This part also underscores the influence of CDK4/6 in G1 phase progression.

Question

In Table 1, I recommend adding subsection headings to better organize the therapeutic strategies presented. For instance, Section A refers to CDK4/6 inhibitors with Fulvestrant and novel SERDs. Adding an appropriate heading to the groupings would add clarity for the reader. Furthermore, please consider referencing the relevant parts of the manuscript where these tables apply.

Answer

As suggested, in Table 1 subsection headings to better organize the therapeutic strategies presented have been added.

Question

Some parts of the manuscript require rephrasing

Answer

As suggested, the phrases that have been appointed have been rephrased. They have been underlined in yellow.

Round 2

Reviewer 1 Report

Comments and Suggestions for Authors

Recommendation:Acceptable,andnoneededtoberevisedgreatlyfurther

Comments:

Inthismanuscript,theauthorsconsideredthat the review article,which the title “

Molecular mechanisms and therapeutic strategies to overcome resistance to endocrine

therapy and CDK4/6 inhibitors in advanced ER+/HER2- breast cancer”After

necessary revisions with the guidience of the reviewer suggestions,This article could be

acceptable,and no needed to be revised greatly further..
